# The 'Arab Clans' Discourse: Narrating Racialization, Kinship, and Crime in the German Media

**Özgür Özvatan** , **Bastian Neuhauser and Gökçe Yurdakul \***

Berlin Institute for Integration and Migration Research, Humboldt-Universität zu Berlin, 10117 Berlin, Germany
\* Correspondence: gokce.yurdakul@sowi.hu-berlin.de

**Abstract:** In the last decade's media discourse, particular Arab immigrant groups received the name 'Arab clans' and have been portrayed as criminal kinship networks irrespective of actual involvement in crime. We question how 'Arab clans' are categorized, criminalized, and racialized in the German media. To answer this question, we collected clan-related mainstream media articles published between 2010 and 2020. Our first-step quantitative topic modeling of 'clan' coverage (n = 23,893) shows that the discourse about 'Arab clans' is situated as the most racialized and criminalized vis-à-vis other 'clan' discourses and is channeled through three macro topics: law and order, family and kinship, and criminal groupness. Second, to explore the deeper meaning of the discourse about 'Arab clans' by juxtaposing corpus linguistics and novel narrative approaches to the discourse-historical approach, we qualitatively analyzed 97 text passages extracted with the keywords in context search (KWIC). Our analysis reveals three prevalent argumentative strategies (Arab clan immigration out of control, Arab clans as enclaves, policing Arab clans) embedded in a media narrative of ethnonational rebirth: a story of Germany's present-day need ('moral panic') to police and repel the threats associated with 'the Arab clan Other' in order for a celebratory return to a nostalgically idealized pre-Arab-immigration social/moral order.

**Keywords:** media; critical discourse analysis; corpus linguistics; racism; crime; immigrants; Germany





## 1. Introduction

Germany is a country with a long and troubled history of immigration. Today, more than 20 million people with migration history live in Germany, with the most well-known group in Germany being the Turkish immigrants, who have been subjected to discrimination and stigmatization for many decades (Yurdakul 2009; Statistisches Bundesamt 2021). German politics has only recently started to acknowledge that Germany is in fact a 'postmigrant' society, centrally defined by the presence and experiences of immigrants, in which migration became a key social policy field through which other social policies (e.g., gender and class politics) were debated (Foroutan 2019). Specifically, after the discovery of the vaccine against COVID-19 by two Turkish immigrants, the successful stories of Turks in Germany became mainstream in the German media and politics. However, other immigrant groups, such as Arabs, became the new target of stigma and discrimination, especially after the rise of the German Alternative für Deutschland, the right-wing populist party, founded in 2013 (Özvatan 2019). In the last decade, we observed that Arab immigrants take on the trope that Turkish immigrants used to have in German media discourses. In this article, we unpack how the German media discourses apply the term 'Arab clans' to refer to the problematized 'Other' in the German media discourse, from 2010 through 2020.

In this study, we aim to show how 'Arab clans' are positioned in relation to other clan-related discourses in the media. Through our data analysis, we provide a detailed and sophisticated explanation of how 'Arab clans' are discursively constructed and embedded in German mainstream media narratives. We believe that this is an important topic for scholars who are interested in immigration and media studies with a specific focus on criminalization of racialized immigrants.

We aim to contribute to the existing literature in three theoretically and methodologically innovative ways. First, we combine the literature of cultural sociology with corpus linguistics methods (Bleich et al. 2018). Second, we draw on the growing literature in the nexus of critical discourse studies, corpus linguistics, racism, and migration research (cf. Baker et al. 2008). Third, we apply a cutting-edge methodological reformulation of the discourse-historical approach that incorporates the concept of narrative (Forchtner 2021). In so doing, we focus on the cultural production of narratives in critical discourse studies (Gavriely-Nuri 2017) and introduce the concept of 'narrative context' with which we develop the methodological relationship between discourse and narrative (Forchtner and Özvatan 2022). There is currently no study in Germany (nor, for that matter, in Europe) that systematically analyzes the narrative and discursive production of 'clan' as a concept that points to organized crime and its association with immigrants' racialized groupness. We aim to fill this gap empirically, theoretically, and methodologically in our study. In so doing, we seek to reveal systemic racialized biases in knowledge production about Arab and Muslim minorities in the German media.

Below, we first discuss our framework for analyzing racism, migration, criminalization, and groupness in the media. We then turn to a description of the German case, providing background on the context for Arabs in Germany. This is then followed by a discussion of our methodological approach which integrates corpus linguistics, critical discourse analysis, and narrative analysis. Through a three-tier analysis, we reveal the narrative foundations of discursive processes around 'Arab clans' and demonstrate how this discursive difference of an 'Arab clan' is embedded in a narrative context, namely in the narrative of German ethnonational rebirth. Finally, we present a short discussion of future research in the light of our findings.

## 2. Narrative Context, Racism, and the Media in Critical Discourse Studies

The racialization of immigrant groups, particularly of Muslims, in Christian-majority Western societies has been observed theoretically (Saeed 2007; Said 1979) and empirically (e.g., Bleich et al. 2015; Bleich and van der Veen 2021). Previous research clearly showed the connection between racialization and criminalization of Muslims, focusing on different media sources, such as TV and newspapers (Farris and Silber Mohamed 2018; Karim 2000; Morey 2011). It is thanks to Cultural Studies, and the postcolonial thinkers therein in particular, that the significance of 'moral panics' has been recognized in the broader social sciences. Cohen ([1972] 2011) and Hall (1978) coined that term vis-à-vis the media and state representations of criminalized and racialized social groups. According to Cohen, moral panics represent situations where 'folk devils' are manufactured when a "condition, episode, person or group of persons emerge to become defined as a threat to societal values and interests" (Cohen [1972] 2011, p. 1) and consequently face ostracism and societal sanctions. Hall puts emphasis on the ways in which this process gives rise to and collectively legitimates state violence: he contends that moral panic "appears to us to be one of the principal forms of ideological consciousness by means of which a 'silent majority' is won over to the support of increasingly coercive measures on the part of the state, and lends its legitimacy to a 'more than usual' exercise of control" (Hall 1978, p. 221). In this article, we show how moral panics are narratively constructed in the German media discourse about 'Arab clans' (see Findings).

The added value of Critical Discourse Studies (CDS) accounts is that they have illuminated how anti-Muslim racism shapes and is shaped by media discourse about Muslims (Cheng 2015; Poole and Richardson 2006; Richardson 2004). Embedding this piece in cultural sociology, CDS facilitates the unveiling of what constitutes the social bond and its critique via the analysis of "the textured webs of social meaning" (Alexander and Smith 2001, p. 137).

We join this scholarship in two innovative ways: First, we use CDS in systematically assessing how current uses of criminalization in the media form part of a racializing process that renders Arab immigrants as not belonging to Western societies such as Germany

(Hamann and Yurdakul 2018). Second, this study focuses on anti-Arab narratives in the German media discourse. We challenge the homogeneous term of "Muslims in Germany" and show how specifically Arab immigrants are racialized and criminalized in the German media vis-à-vis other criminalized groups.

In relation to the juxtaposing of racialization and criminalization in and through discourse, and understanding the patterned relationship between text and culture, we firstly draw on the Discourse-Historical Approach (DHA) within Critical Discourse Studies (CDS). The DHA considers discourse to be a highly contextual cluster of semiotic practices in a dialectical relationship with its social environment, and as such is strongly focused on contextualization. It differentiates various discursive strategies as a "more or less intentional plan of practices (including discursive practices) adopted to achieve a particular social, political, psychological or linguistic goal" (Reisigl and Wodak 2009, p. 94). Each strategy relates to a specific heuristic question aimed at unearthing the power structures behind discourse. The strategy of nomination asks how persons, objects, events, processes, and actions are referred to linguistically. The strategy of predication inquires which characteristics, qualities, and features are attributed to social actors, objects, events, and processes. Thirdly, by looking into which arguments are activated in a given discourse, we analyze the strategy of argumentation. In perspectivation, it is inquired from which viewpoint these strategies are expressed and which position the writer/speaker takes on in their formulation. Mitigation/intensification explores whether utterances are expressed overtly and whether they are attenuated or reinforced (Reisigl and Wodak 2009).

Secondly, we theorize the media construction of 'Arab clans' by drawing on the theoretical framework of narrative context. In ontological terms, we recognize the fundamental role of narrative (Somers 1994) and, more specifically, of narrative genre, in providing narratively structured contexts (lifeworlds) for communicative action. This overarching structure of cultural meaning we define as narrative context. Consequently, we follow Forchtner and Özvatan's argument (2022) that (1) it is within a specific (socio-historical) narrative context that discursive practices unfold their meaning, and (2) that the narrative context is itself structured into four narrative genres which shape discourse.

Following Frye (1957) and White's (1973) generic understanding of narrative genres (i.e., narrative forms), a romance's plot line leads steadily towards a happy ending. V-type comic plots are characterized by a continuous fall (first half of 'V') which is followed by a sharp ascend to a happy ending (second half of 'V'). Both narrative genres' resolutions construct a black-and-white discursive space, that is, arguments raised by actors signified as 'villainous' are preemptively discarded. Hence, the moralization and polarization of 'the good' and 'the bad' are realized by the interpellation of social roles through romantic and comic emplotment. Tragedy and irony, however, acknowledge villainy and failures within the set of 'the good us'. Tragedy's plot line steadily declines into a cathartic ending which facilitates the collective decentering and distancing from previously held convictions. Similarly, the ironic plot is defined as a radical mockery of romance, which means that 'our' previously held convictions are mocked and reconsidered. Irony, thus, mirrors a mechanism of self-reflexive decentering (instead of mocking 'others') while there is no common understanding about irony's plot line in the literature.

Amalgamating narrativity and CDS, we join arguments that the discourse-historical approach needs to install narrative (genre) into its conceptual architecture (Forchtner and Özvatan 2022; Forchtner 2021) because, again, narrative is the principal way "our species organizes its understanding of time" (Abbott 2008, p. 3). Forchtner (2021) has demonstrated that the four discourse-shaping, archetypal narrative genres in Western societies constrain (romance and comedy) or trigger (tragedy and irony) the inclusion of 'Others' by the interpellation of social roles (e.g., heroi/nes, victims, bystanders, helpers, and villains). As such, these configurations of social roles draw social boundaries, they delimit or expand the set of legitimate actors, perspectives, and arguments in a discourse.

Within CDS, the analysis of racist and anti-immigrant discourses in the media builds particularly on van Dijk's scholarship. In an extension of his earlier works on media

portrayals of ethnic minorities, van Dijk (1991) demonstrated that in Western societies it is through text and talk in the media that racial signification and white dominance is (re)produced. This finding has generated a body of research (in CDS) into the many-faceted ways that (neo)racism against a diverse set of racialized and 'migranticized' groups is mediated in public communication (Breazu and Machin 2019; Dahinden 2016; Del-Teso-Craviotto 2009; KhosraviNik 2010; Teo 2000; Wodak and Matouschek 1993).

### 3. The Immigration Context of Arabs and 'Arab Clans' in Germany

In Germany, many Arab immigrants arrived in the 1960s predominantly from Morocco and Tunisia on the basis of guest worker contracts. After legislation regulating family reunification was introduced in the early 1970s, immigrants began to bring their families to Germany. Triggered by the Lebanese Civil War and the Arab–Israeli conflict, the 1970s also saw a growing number of Lebanese and de facto stateless Palestinian asylum seekers in Germany. Since then, the largest wave of Arab immigration came in 2015 with the predominantly Syrian refugee influx during and after the so-called 'long summer of migration' (Hamann and Yurdakul 2018). According to official immigration statistics collected in 2020, there are more than 1.4 million Arab immigrants in Germany today without German citizenship (Statistisches Bundesamt 2021). This statistic does not include Arabs who have received German citizenship or Germans of Arab descent, because Germany has not collected general statistics on ethnic background since, and because of, the Shoah. According to a representative study, immigrants from Arab countries (and Turkey) feel most discriminated against by German institutions and government offices (Brücker et al. 2014).

Although Arab immigrants densely populate certain urban areas in German cities, and form a visible minority in Germany, there is limited research about that differentiation within Arab communities. In the German media, they are usually discussed under the general label 'Muslims'. Lebanese and Palestinian communities represent only a comparatively small fraction of the Arab population in Germany compared to other immigrant groups, such as Iraqis and Syrians, while it is argued that they are subjects of a 'public gaze' (Atshan and Galor 2020). Specifically in the German media, *arabische Großfamilien* (Arab extended families) and *arabische Clans* (Arab clans) are used to refer to Lebanese and Palestinian families who carry notorious family names such as Abou-Chaker, Al-Zain, Chahrour, Miri, and Remmo. Qualitative research indicates that these terms have become synonymous with racialized organized crime in Germany (Ghadban 2016; Jaraba 2021; Rohde et al. 2019). At the same time, most studies on criminalization in Germany stigmatize Arab populations, evoke moral panic, support persecution as the initial act (Boettner and Schweitzer 2020; Feltes and Rauls 2020), and only rarely recommend prevention (Dienstbühl 2020).

Our research in the newspaper databases shows that the term 'clan' started to circulate in the media relatively recently. The trigger to use the term 'clans' in the media or in politics is not traceable, but we assume that refugee settlements from Syria and Iraq after 2015 might have played a role in increasing stigmatization. We chose the year 2010 as a starting point for data collection because the term 'Arab clans' started appearing in the news only then. The term 'clan' found its way into government documentation for the first time through a 2018 national situation report on organized crime commissioned by the Federal Criminal Police Office (Bundeskriminalamt 2019). In the report, the word Clankriminalität (clan criminality) was defined as a branch of organized crime conducted by "criminal members of ethnically isolated subcultures" (Bundeskriminalamt 2019, p. 28) characterized by patriarchal-hierarchical family relationships, common ethnic descent, their own set of values, a lack of willingness to integrate into the majority society, a fundamental rejection of the German legal order, and obstruction of criminal investigations by German law enforcement. The report further points out that because twenty-four out of forty-five registered "organized crime groups" are defined as being "of Arab descent" (*arabischstämmig*), specific weight is given to organized crime conducted by these groups. Although the report covers the ethnic diversity of these social groups (Bundeskriminalamt

2019, p. 33), including not only those of Turkish background, but also Germans and others, they are nevertheless merged as 'Arab clans'. Given that the racialized and migranticized definition of 'clan criminality' used in the report has also been used in some federal states and at the national level in parliamentary processes, the conceptualization and definition of 'clan criminality' by the Federal Criminal Police Office can be considered a key site for governmental sense-making about organized crime in Germany and resembles aforementioned 'moral panics'.

## 4. Data and Method

Methodologically, we integrate both quantitative and qualitative methods using Corpus Linguistics (hereafter CL) and CDS. Over the past two decades, these approaches have seen mounting integration (Fairclough et al. 2007) and together form productive synergies for sociological analysis and 'social critique' (Baker et al. 2008). CL refers to the computer-assisted empirical analysis of language through the use of a large corpora of texts, while CDS analyzes discourse as a "socially constituted as well as constitutive semiotic practice" realized as "a communicative and interactional macrounit that transcends the unit of a single text or conversation" (Reisigl and Wodak 2001, p. 45). As such, corpus linguistics can serve as a means for providing "a general 'pattern map' of the data" (Baker et al. 2008, p. 295) which can then be further analyzed empirically in-depth with CDS approaches. While we abstain from the integration of Computational Text Analysis (CTA), and Computational Grounded Theory (Nelson 2020) in particular, we follow Nelson's claim that topic modeling is a 'natural tool' for sociologists of discourse and culture because of "the efficiency of topic modeling algorithms, the easily interpretable results, and the justifiable assumptions built into the algorithms" (Nelson 2020, p. 17). In fact, we use CL (i.e., topic modeling) as a methodological entry point as recommended by Partington (2003) for (1) descriptive analysis, (2) case selection purposes, and (3) keywords in context (KWIC) searches in the context of sampling text passages for in-depth analysis.

We gathered data for 'clans' in general instead of 'Arab clans' in order to assess the variety of 'clan' discourses in German media and to then situate the discourse about 'Arab clans' within that discursive arena. This strategy allowed us to show that 'Arab clans' are more racialized and criminalized in comparison to other 'clans' in the media based on the topic model outputs for each 'clan discourse'. In the further steps of thicker description (discourse-historical and narrative analysis), we focused *solely* on the discourse about 'Arab clans' to illuminate the discursive and narrative mechanisms through which the racialization and criminalization of 'Arab clans' is linguistically realized at a deeper level of meaning. The representative sample of quality German media outlets comprises a total of 23,893 media articles resulting from three distinct search terms: "*clan\**",[1] "*großfam\**" (short for "extended family"), and "*gangs*" to control for the use of alternative terms. Articles were retrieved from eight mainstream German newspapers distributed along the left–right continuum, accessed and downloaded through the NexisUni and Factiva databases (see Table 1). The available articles were sorted by date and converted into Rich-Text (.rtf) format to enable further processing with Python. We chose the year 2010 as a starting point for data collection due to the fact that the 'clan' terminology increasingly started appearing in the news as was shown in prior explorative analyses. The end point for the data collection was set at 31 December 2019, which is when data collection began. The availability of the media archives in the Factiva or Nexis Uni databases did not affect the choice of the media outlets used in the analysis with the exception of Frankfurter Allgemeine Zeitung (FAZ), whose archives were not accessible for the researchers at the time.

**Table 1.** German media sources for data collection (1 January 2010–31 December 2019).

| Media Source | Political Leaning | Number of Articles | Database |
|---|---|---|---|
| Bild | Right (tabloid) | 1534 | NexisUni |
| Die Zeit | Center-left | 1620 | NexisUni |
| Süddeutsche Zeitung (SZ) | Center-left | 7849 | Factiva |
| Die Tageszeitung (taz) | Left | 2063 | NexisUni |
| Die Welt | Center-right | 7402 | NexisUni |
| Stern | Center-left | 445 | NexisUni |
| Der Spiegel | Center-left | 2480 | NexisUni |
| Focus | Center-right | 500 | NexisUni |
| Total | | 23,893 | |

Drawing on our novel approach to DHA (Forchtner and Özvatan 2022), we offer three levels of analysis. First, we use corpus linguistics to present a quantitative entry-level *topic model analysis* of macro topics characterizing the discourse about 'clans' and its synonyms. To this end, we built on an open-source software in Python code developed by Bleich and van der Veen (2021). As a first step, the downloaded articles were sorted and deduplicated, then cleared of punctuation and stop words. After defining minimum occurrence thresholds and maximum proportions, we identified a total of 364,747 relevant words ('terms') from the overall text corpus. With the help of algorithmic non-negative matrix factorization (NMF), we then aggregated these terms into 24 topics characterized by tested internal coherence. Based on this process of topic modeling, we created an intertopic distance map using pyLDAvis to visualize the salience of each topic in the corpus through size and the relationships between topics through distance and proximity (Chuang et al. 2012). Second, to assess the deeper levels of social meaning through thicker description of the discourse about 'Arab clans,' we performed a qualitative, in-depth *discursive strategies analysis* of the discourse about 'Arab clans' allowing us to illustrate text-level linguistic realizations of the identified implicit narrative genre. To this end, we used the five most salient terms of six topics appearing in the 'Arab clans' cluster as keywords with which to identify text passages with outstanding analytical relevance in our corpus.[2] Eradicating doubles and filler terms, we derived 27 especially salient terms from the corpus which we used as a basis for a KWIC search. In this process, we extracted 97 text passages containing 8590 words, which we then coded using MAXQDA. We coded the material for the four discursive strategies nomination, predication, argumentation, and perspectivation—the fifth strategy, mitigation, we ignored because we did not expect any substantial added value. Third, we conducted an interpretive *narrative genre analysis* of the discourse about 'Arab clans' as a way to identify the dominant media narrative, which preconfigures the discursive realizations identified in level 1 and 2. It is only by layering these three analytical methodologies—quantitative entry-level topic modeling, qualitative in-depth discursive strategy analysis, and interpretive narrative genre analysis—that we may arrive at theorizing media portrayals in the discourse about 'Arab clans'.

## 5. Findings

### 5.1. Level 1: Topic Modeling

Topic modeling can serve as an effective way to identify macro-patterns in the corpus. Using the intertopic distance map, we bundled the resulting twenty-four topic models identified through CL into four 'clan' discourse clusters: MENA (Middle East North Africa), pop culture, car industry, and Arab clans. When factoring out topic 1, which features analytically irrelevant filler terms (57.4% of all terms), the MENA cluster accounts for 19.7%, the pop culture cluster for 13.4%, the car industry cluster for 2.6%, and the Arab clans cluster for 44.4% of all analytically meaningful terms.

Making use of the intertopic distance map, which is a comparative visualization of the topic model output (see Section 4), we present a model (see Figure 1) which situates these four resulting clan discourses vis-à-vis their degree of racialization and criminalization.

That is, the intertopic distance map emerged as a topic modeling output, after which we interpreted all topic models to identify all four 'clan' discourses and, eventually, we ended up defining both axes of the presented model: racialization and criminalization. By racialization we mean signification processes of social groups which rely on ancestral, faith-based, 'racial', ethnic, and cultural markers. Criminalization reflects references to supposed and actual crime and violence.

Following our interpretation of the topic model outputs, one resulting clan discourse appears as racializing but not criminalizing (MENA, predominantly covering political developments in that region), another appears as scarcely criminalizing and racializing (pop culture, covering public figures such as 'the Kardashian clan'), and a third appears as less racializing but strongly criminalizing (car industry, covering 'clan criminality', for instance, in the Volkswagen takeover battle between the Piëch and Porsche families). The discourse about Arab clans occupies the intersection between the criminalizing and racializing topic model outputs, depicted in the bottom right corner of Figure 1.

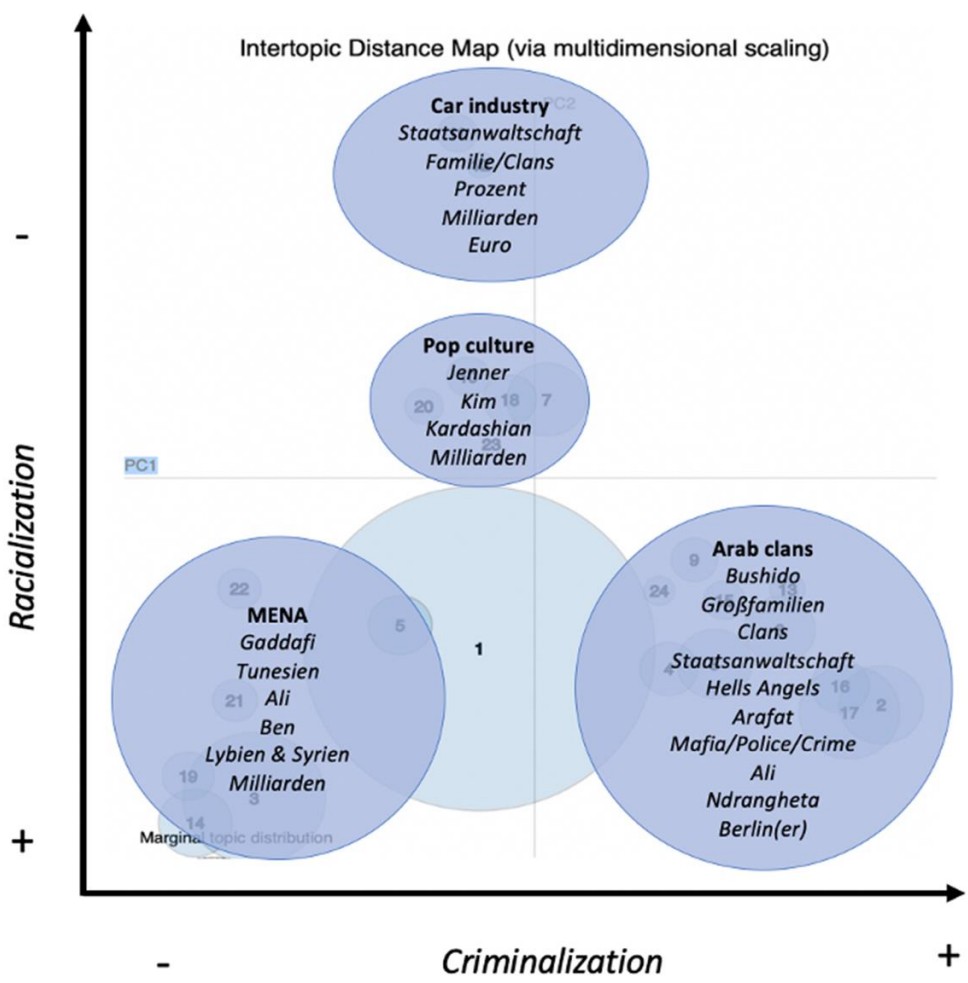

**Figure 1.** Intertopic distance map of the German media discourse about 'clans' (2010–2019)[3].

Examining the most salient terms within each topic in the 'Arab clans' cluster—Bushido, *Großfamilien* (extended family), clans, *Staatsanwaltschaft* (state attorney), Hells Angels, Arafat, Mafia, police, crime, Ali, 'Ndrangheta, Berlin, and Berliner—and qualitatively interpreting each topic model output individually, we identified three macro topics that constitute the discourse about 'Arab clans': family and kinship, law and order, and criminal groupness (see Figure 2), which in quantitative corpus linguistic terms already indicate a discursive construction of a moral panic, i.e., the ascription of 'folk devils' and their need to be policed.

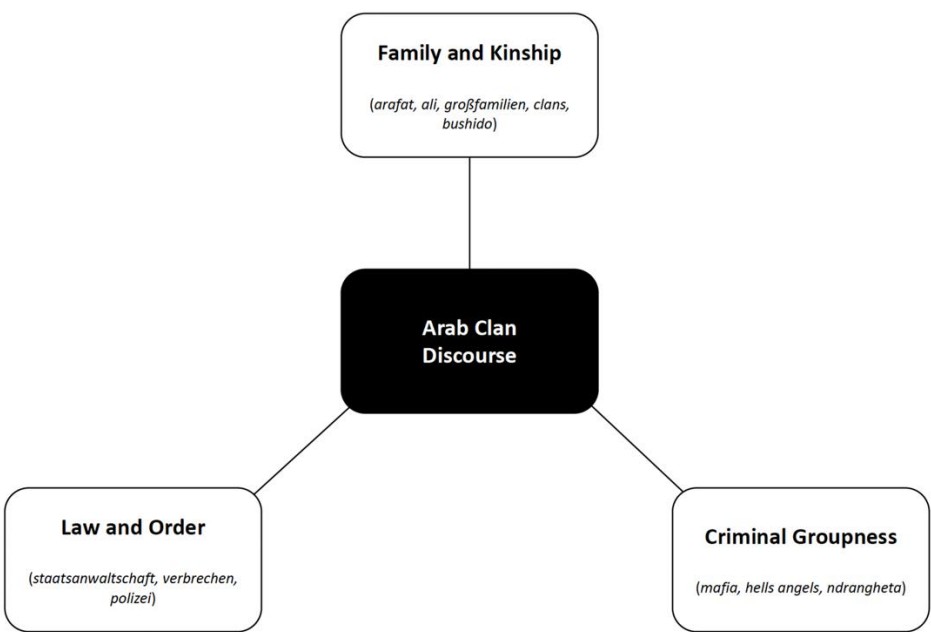

**Figure 2.** Macro topics that constitute the German media discourse about 'Arab clans' (2010–2019).

Figure 2 illustrates the three macro topics that constitute the German media discourse about 'Arab clans' between 2010 and 2020. The discourse about 'Arab clans' revolves first around the macro topic of family and kinship. The corresponding topic model outputs for this macro topic indicate that the discourse about 'Arab clans' is familialized through the semantic practice of naming individual 'clan' members, as well as through use of the word *Großfamilie* (extended family). In other words, the family and kinship macro topic constitutes an understanding of 'Arab clans' as people organized into close and extensive social networks of kinship ties. The second macro topic, criminal groupness, is realized through the practice of referencing the Mafia and other organized criminal groups when depicting 'Arab clans' such as 'Ndrangheta' and the Hells Angels. The third macro topic, law and order, associates 'Arab clans' with the intersection of state authorities and policing. It is through these three macro topics (family and kinship, criminal groupness, law and order) that the discourse about 'Arab clans' is discursively constituted.

*5.2. Level 2: Discursive Strategies Analysis*

In the second phase of the research, we utilized a discourse-historical approach to conduct an in-depth discursive strategies-level analysis of the extracted ninety-seven text passages. We turned to discourse-historical approach's analysis of discursive strategies (Reisigl 2017) as a way of elucidating the linguistic realization of the discourse about 'Arab clans'. Table 2 illustrates how each discursive strategy feeds into the construction of racialized groupness and criminalization in the discourse about 'Arab clans'. Our coding reveals that media uses of the term 'Arab clan' relate to criminalization, racialization, and kinship. 'Arab clans' and individual clan members are referred to as 'criminals'; 'crime' is most often supplemented by 'organized' in combination with family names ('Abou-Chaker' and 'Miri'), their racialized backgrounds ('Lebanese' and/or 'Arab'), and with the expression 'extended family', which again implies the making of a moral panic around 'Arab clans' and their alleged threat to the German state.

We present three dominant argumentative strategies through which its key narrative, the promise of ethnonational rebirth for non-Arab Germans once we policed 'them' (see Figure 3), is linguistically realized.

**Table 2.** Discursive strategies within the German media discourse about 'Arab clans' (2010–2019).

| Strategy | Objective | Examples |
|---|---|---|
| **Nomination** | Naming of protagonists: actors, objects, phenomena, events, and processes | Crime/criminal, (Ibrahim) Miri, Bushido, (Arafat) Abou-Chaker, extended family (*"Großfamilie"*), Lebanese and Arab |
| **Predication** | Features of naming: realized by evaluative, stereotypical attributions of positive/negative traits or by references to them | Out of 167 predications, not a single feature of naming decoupled from organized crime, criminality, violence, sexual violence, (clan) brutality, fraud, robbery |
| **Argumentation** | Justification of features: arguments employed (supposed) to persuade the addressee(s) of the validity, truthfulness, and accuracy of the claim | *Migration* (n = 34): Arab clans undermine the German immigration and asylum system<br>*Law and order* (n = 49): Arab clans undermine German law and order<br>*Organized crime* (n = 25): Arab clans are defined as large families in which "group" crime is organized<br>*Gendered clan socialization* (n = 13): Arab clans socialize their members into archaic gender roles (i.e., archaic masculinity)<br>*Clan brutality* (n = 26): Arab clans form "groups" in which brutality against Others is legitimatized<br>*Parallel social order* (n = 22): Arab clans create a parallel social order beyond German law and order |
| **Perspectivation** | Strategies of distancing and/or involvement of the speaker's position | The voices of authors, scientists, experts, and intellectuals are mostly absent (n = 2)<br>The protagonists/subjects of the 'Arab clan' discourse are silent/silenced (n = 4)<br>The primary discursive agents in media reports are state (security) and political authorities (n = 49) |

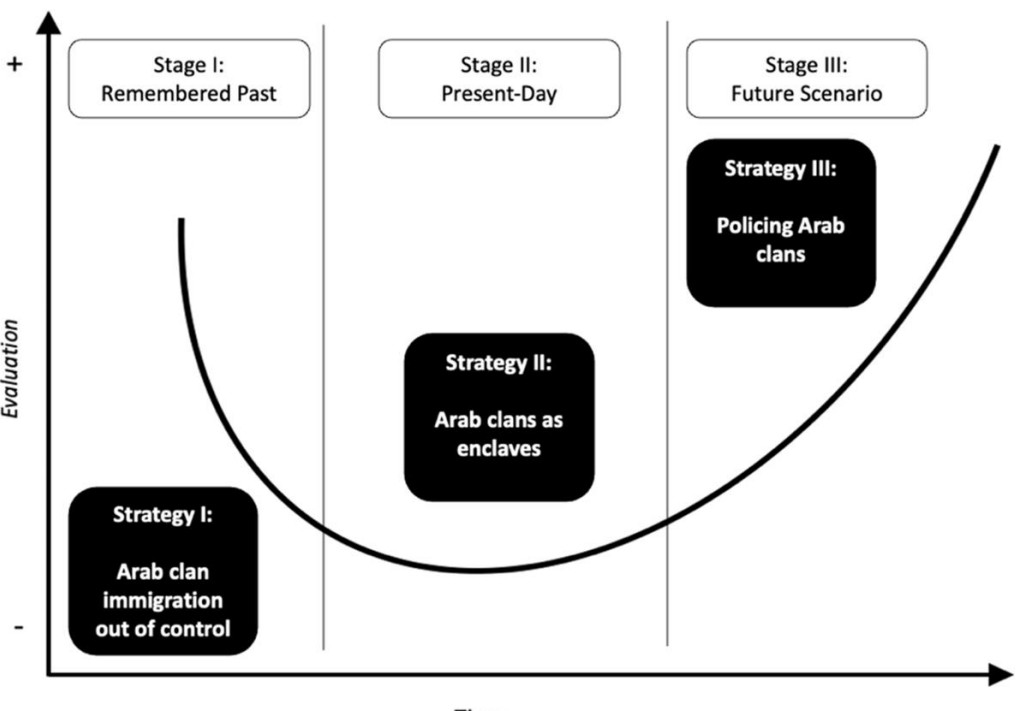

**Figure 3.** Integrated representation of argumentative strategies used in the comedy-romance emplotment of German media discourse about 'Arab clans' (2010–2019).

'Bushido' is another frequent name associated with nomination and predication strategies deployed in relation to the 'criminal Abou-Chaker clan'. In fact, the predication strategies lend evidential support to the idea that there is not a single nomination of 'Arab

clans' unrelated to crime. One might argue that this is unsurprising given that 'clan' is (allegedly) semantically closely tied to crime. Yet Figure 1 demonstrates that non-criminalizing notions of 'clans' exist (see the entries under pop culture).

These nomination and predication strategies of the discourse about 'Arab clans' shed light on the discursive construction of a criminal 'Arab Other'. Against the background of the V-type comic emplotment, such nomination and predication strategies feed into the macro topics of law and order, criminal groupness, and family and kinship to create feelings of threat posed by 'them'. The threat that 'they' pose is further disseminated through the use of certain perspectivation strategies. Specifically, involvement is created by citing state (security) authorities and politicians who are staged as authentic sources who 'prove' the trouble and chaos caused by 'Arab clans', which confirms existing scholarship in CDS on the question of who and what is treated as 'authority' (van Leeuwen 2007; Wodak 2015). Further, this confirms traditional knowledge in media studies that sources, especially state authorities, 'make the news' (Sigal 1986).

Concerning less privileged and talked-about subjects of media discourses, Korteweg and Yurdakul (2014) show that veiled women became visible interlocutors in the headscarf debates after the debate was initiated by privileged actors (e.g., majority society actors), while it is striking that the 'Arab clan' subject is allocated no single speech act by the media in our study (see Table 2).

We identified six argumentation strategies within the discourse about 'Arab clans': migration; law and order; organized crime; gendered clan socialization; clan brutality; and parallel social order (see argumentation in Table 2). We grouped these six argumentation strategies into the following three overarching argumentation strategies according to their interrelatedness: (1) Arab clan immigration out of control; (2) Arab clans as enclaves characterized by a parallel social order, groupness, and gendered clan socialization; and (3) policing Arab clans (Figure 3). It stands out that all three relate to immigration, racialization, and policing, another empirical indication that a moral panic was in the making by the media in that discourse.

5.2.1. Argumentation Strategy I: Arab Clan Immigration out of Control

This discursive intervention postulates that the 'uncontrolled' immigration of 'Arab clans' to Germany forms an existential threat to German ethnonational identity and order. In excerpt 1, from an article published on Welt Online in 2019, it is claimed that the members of the 'Arab clans' are able to easily and illegally (re)immigrate to Germany. Here, it is referred to one member of the 'Miri clan' who was deported after a series of criminal activities, but managed to re-enter Germany by the use of different personal identities and continued committing crimes:

> After his deportation to Lebanon, a leading member of the Lebanese Miri clan returned to Bremen. This was confirmed by the State Ministry of the Interior on Wednesday. Entry and residence were refused. Apparently, entry took place illegally. Criminal investigations on this matter have been initiated. (Welt Online 2019a)

In excerpt 2, drawing from a 2019 Bild article, the discursive construction of untrustworthy and unchanging criminal 'Arab clans' is rigidified. The implicit message is that 'they' do not learn from mistakes and following punishments. Instead, 'Arab clan' members try to trick the system by complicating their migration history and appealing to several institutional frameworks at the same time (finding legal loopholes), thus delaying deportations. Coverage of the 'Miri clan' tricking the German asylum system was a frequent topic in the KWIC dataset.

> Miri applied for asylum as a 13-year-old in 1986. His [present] application can therefore be treated as a follow-up application. Because follow-up applications are more complicated to decide upon, refused asylum seekers often use them to delay their deportation . . . In one instance he claims to be born in 1980, in

another in 1982, in yet another in 1984, sometimes in Beirut (Lebanon), sometimes in Turkish Mardin. (Göke and Uhlenbroich 2019)

In excerpt 3, from an article published in Bild in 2019, mobilization against 'Arab clans' is organized. Party-political actors emerge as authorities who demand strict state responses to protect the taxpayers from economic harm by the asylum system. They call for a stricter asylum policy.

[This is] (a) horror scenario for Christoph de Vries (44) of the CDU, the expert from the Federal Ministry for the Interior: "We have to ask ourselves whether we want to continue to finance an asylum system where criminals ignore bans on entry and residence, enter the country illegally, and, on top of everything else, are able to apply for an asylum that protects them from deportation until decided".: "We must limit significantly the possibility of duplicate applications for asylum". Support was voiced by the opposition. The Secretary General for the FDP, Linda Teuteberg (38) to Bild: "The law has to be enforced, and the proceedings must be sped up. (Bittner et al. 2019)

By generalizing control strategies, this discursive strategy (Arab clan immigration out of control) casts generalized suspicion on the social activities of all members of the Arab minority, creating feelings of loss of control, and thereby serves justifying content for the need to restore it by strict(er) and more effective policing measures of Arab immigration to Germany. Its underlying reasoning further iterates that once immigrants are deported for 'good' reasons (for example, for being persistent offenders), 'we' should introduce strict generalized control mechanisms that guarantee prevention of these groups reentering 'our' borders. These discursive interventions create an image of 'Arab clan' members as untrustworthy and non-learning subjects—a (supposed) antidote to civilized, rational, and learning German subjectivities.

State authorities and politicians serve as legitimizing voices, both confirming and justifying argumentative claims through this particular 'state positionality'. Consequently, this discursive strategy feeds into both the law and order and the criminal groupness macro topics.

### 5.2.2. Argumentation Strategy II: Arab Clans as Enclaves

This discursive strategy involves interventions that presume 'Arab clans' to be paramount actors in the construction of immigrants' 'parallel societies' in Germany. It is within 'Arab clans', it is reasoned, that socialization into 'monist' ethno-religious moral orders is organized within sealed social group boundaries (Lamont et al. 2016) and the German state's order and authority is delegitimized. In excerpt 4, taken from a 2015 essay published by senior SPD member Peer Steinbrück in Die Zeit, the claim that these "parallel societies" have been taken over by "groups with migration background" utilizes a balanced nomination strategy whereby the alleged threatening immigrant "groups" and "the organized criminality of clans" are depicted as spreading across Germany. This reiteration depicts the threat as neither static nor harmless but as a "creeping" threat. The assertion that 'we' are facing a "creeping" threat is reinforced via reference to the legitimizing power of a blurry people—namely, the "many citizens" (a proxy for white German citizens) who allegedly are the 'true' owners of the nation and the affected neighborhoods. It is this fuzzy people that is said to "perceive a creeping takeover" by the immigrant criminal, non-German groups.

Many citizens perceive a creeping takeover of neighborhoods by groups with migration background and the organized criminality of clans. (Steinbrück 2015)

In excerpt 5, from an article published in Die Zeit in 2013, the "creeping takeover" is portrayed as impervious to 'external' permeation by the state and other German host society institutions. Legitimacy is once again derived from a confirmative and blurry collective body, this time via reference to "asking around the streets of Neukölln", a neighborhood of Berlin known for its contemporary high visibility of immigrant life and working-class trajectory from past to present:

> Asking around the streets of Neukölln, the Abou-Chakers appear as a mix between ordering power, raiding party, and family enterprise. They sometimes arrived in their BMWs to settle disputes, as though they were sheriffs. There is rumor of "protection money". And that one can borrow money, under horrendous conditions. It'll hurt if you can't pay it back. State's Attorney Kamstra says, "Some clan members have tried to pervert or buy off testimonies. (Musharbash 2013)

In excerpt 6, from Der Spiegel in 2010, the state's judicial and executive powers are delegitimized in these social networks, with the example of the Abou-Chaker clan in Berlin's Neukölln neighborhood offered as proof. Blood relations are presented as serving the glue of such social networks' impermeable groupness in which children "grow up largely unsupervised in these criminal structures" and the state is said to have "no chance of reaching these families". Within such sealed networks, the descendants of 'Arab clans' are socialized into family cultures ("criminal structures") dominated by crime, violence, and sexist masculinity. Socialization into clan criminality is presented as a gendered phenomenon, with 'Arab clan' women engaging in 'soft', less physically violent crime and men in 'hard' crime involving physical and symbolic violence:

> Generally speaking, an overview of the criminal history of some clans demonstrates that female family members prefer to steal, while male crimes touch on all areas of the criminal code, including drug and property offenses, verbal abuse, threats, robbery, blackmail, grievous bodily harm, sexual misconduct, pimping, all the way to murder. Children grow up largely unsupervised in these criminal structures. The state has no chance of reaching these families. (Heisig 2010)

These excerpts illustrate how the V-type emplotment of a restoration of a pre-existing white Christian German social/moral order is realized. Both the state and the host society are presented as helpless in preempting gendered and criminal socialization into 'Arab clan' groupness.

In sum, the media intervenes to iterate the "creeping takeover" of neighborhoods depicted as being governed by 'Arab clans'. This intervention is legitimized and enforced via reference to state security authorities and politicians and to the broader public ("many citizens" in excerpt 4 and "asking around the streets of Neukölln" in excerpt 5), which jointly serve to signify the present-day catastrophic decline and fall of the German social/moral order caused by 'Arab clans'.

### 5.2.3. Argumentation Strategy III: Policing Arab Clans

This discursive strategy posits that 'Arab clans' undermine the German legal system, giving rise to an intervention into the discourse about 'Arab clans' that argues in graphic detail for a restrictive policing of 'them'. Once again, the discourse entails situating the voices of selected state authorities as authentic and legitimate experts who demand action toward restrictive policing of 'Arab clans'. As may be seen in excerpt 7 from Welt Online, one text-level specific feature of policing-oriented interventions is that the number, magnitude, and brutality of 'Arab clan' criminality is repeated in headlines and text and is described in detail:

> Young man partly scalped: trial against 13 accused in Essen (DPA/LNW): The victim was partly scalped and critically wounded by stabbing. Eight months after the brutal crime, the trial against 13 members of a Syrian extended family has begun at the Essen District Court. (Welt Online 2019b)

The underlying claim reinforces the impression that even though the threat is pressing in both number and magnitude, state authorities have thus far failed to counter such incidents, except those selected as authentic voices. In excerpt 8 from an article published on Welt Online in 2019, the discursive strategy of singling out officials from the Federal Ministry of the Interior who might reverse the increase in such incidents by restrictively policing 'Arab clans' is used. One such prominent figure is North Rhine-Westphalia's

State Minister for the Interior, Herbert Reul (CDU), a well-known proponent of strict law-and-order policies who introduced hard pushes against 'Arab clans' in the recent years. To stress the extent of the threat posed by 'Arab clans', the discursive strategy implies first a narrativization pointing out how in the past "the state did not intervene when they constructed their criminal structures", a fact already established as initiating change for a better future, and, second, by referencing once again the collective "many citizens" said to have lost faith in the ability of state authorities to address socio-political grievances:

> ... but then the state did not intervene when they ['Arab clans'] built their criminal structures. The raids are meant to send an additional signal to the Lebanese community. 'I make trouble so as to compel youth and especially women to consider whether it would be smarter to take a different path,' says Reul Sunday morning in Essen. In Germany, one can 'be economically successful without committing crimes'. The operations are also meant to demonstrate strength outwardly, because many citizens have lost confidence and doubt the authority of the state. And because the police face aggressions. (Frigelj 2019)

Another striking aspect of the policing strategy is marked by its integration-oriented justification. Vulnerable groups ("youth and especially women") are selected as being in need of rescue from these 'Arab clan enclaves'. It is in this way that the call for policing moves beyond the need to secure ethnonational Germans' social/moral order to include calls for policing youth and women's moral integrity via the concept of vulnerability. This realizes Spivak's claim that gendered racialized structures are reproduced by "white men saving ... brown women from brown men" (Spivak 2008, p. 78). This is an instance that indicates that it is through the concept of vulnerability that specific groups of the 'othered' are staged as innocent and in need of a savior, a narrative configuration that helps to reassure 'us' of 'our' moral superiority.

Minister Reul is presented as one such idealized heroic white male figure not only because it is him who is courageous enough to cause "trouble" for these 'Arab clans'. He claims that the state must fight back against 'Arab clans' because "the police face aggressions" and "many citizens have lost confidence and doubt the authority of the state". Moreover, Reul's heroic portrayals in the media are associated with success stories around restrictive policing. In excerpt 9, this success is discursively evidenced by pointing to his approach's spread across Germany. Berlin, where clan criminality is described as "a major issue as well", plans to copy Reul's restrictive "pressure on organized crime". Berlin's State Minister for the Interior, Andreas Geisel (SPD), is singled out and goes so far to even to consider deportations of apprehended 'Arab clan' members.

> Andreas Geisel (SPD), State Minister for the Interior of Berlin, where clan criminality is a major issue as well, welcomed that move. Consequently, the new directives will be followed in Berlin, and the "pressure on organized crime" will be kept up, Geisel announced on Thursday. "Where possible, deportations will be considered". (Welt Online 2019c)

Policing strategies constitute another discursive realization of present-day decadence and, in response to decay, singles out selected state authorities as rare heroic figures who revolt against decline and free-fall. This argumentative strategy makes urgent appeals for the policing of 'Arab clans' and marks an intervention into the discourse about 'Arab clans' that becomes meaningful against the background of the discourse's dominant V-type emplotment: either the plot descends further into a past-Arab-immigration-caused catastrophic dissolution of 'our' social/moral order, or we follow the proposed 'fresh' impetus by 'courageous' white German heroes—presented by the media and supported by voices of justifying state authorities—to avert the fall and ascend to a rebirth of 'our' moral compass and social order by symbolically and socially policing and even excluding 'them', where necessary.

*5.3. Level 3: Narrative Genre Analysis*

In a third and final step, we reconstructed the dominant narrative genre present in the corpus by identifying key characteristics, including events, social roles, and the plot structure. Following this template, we identified a narrative of ethnonational rebirth in the discourse about 'Arab clans' which derives its mobilizing power by the construction of a moral panic in the middle part of the story which promises ethnonational rebirth, if the racialized threat by 'Arab clans' is overcome (Figure 3). This plot presents two antagonistic subjects: the 'peaceful ethnonational German hero/ine' is realized as an antidote to the 'villainous Arab Other' who, as a non-belonging Muslim immigrant to this country, organizes crime in and through racialized kinship ties (clans), threatening the peaceful social/moral order and requiring strict policing by the German state.

Empirically, media portrayals foreground present-day descriptions, whereas presentations of the past and future are rather backgrounded and implicit. While narrative genre analysis follows an interpretive methodology, the degree of interpretation varies. In fact, stages 1 and 3 of the plot (past and future) are rather implicitly asserted, while the empirical presentation of present-day scenarios in stage 2 is less interpretive.

5.3.1. Stage 1: The Selectively Remembered Past of Media Interventions into the Discourse about 'Arab Clans'

Stretching from the selectively remembered past to the present-day decline, discursive realizations in argumentative strategy I (Arab clan immigration out of control) unfold their meaning specifically in presentations of the past in the plot. It delivers an argumentatively packaged reason for 'our' alleged decline from past to present, namely, that 'we' failed to prevent 'our' borders from the reimmigration of certain members of 'Arab clans' after 'we' had deported them.

The media thus narrates a pre-Arab-immigration past idealized as a time of social cohesion and peace. In fact, 'Arab clans' are presented as invaders of German society who disregard German culture and law. Thus, in stage 1 of the plot, a two-layered narrative delegitimation (Author 2022) through the allocation of villainy is present: firstly, and more explicitly, 'Arab clans' are portrayed as lying, violent, and criminal immigrants, and, secondly, and more implicitly, German liberals are presented as those who caused that downfall because they and their 'multicultural' policy-making were naïve.

5.3.2. Stage 2: Portrayals of Present-Day Catastrophe and 'Moral Panic' in Media Interventions into the Discourse about 'Arab Clans'

The depiction of 'our' alleged failures—too-liberal reimmigration stances and policies—feeds into a portrayal of present-day chaos due to too-liberal immigrant integration policies in stage 2. Argumentative strategy II (Arab clans as enclaves) becomes meaningful and delivers reasons for the plot's crucial turning point. The media intervenes in the discourse with the above-mentioned moral panic and villainous portrayals of 'Arab clans' to wake ethnonational Germans up to change the course of the plot: 'we' must unite under the umbrella of restrictive law-and-order policies toward 'Arab clans' so as to restore 'our' social/moral order.

In this V-type emplotment, this discursive strategy depicts a present-day situation of degradation supported by a hyperbolic portrayal of the threat from ruthless 'Arab clans'. In terms of the generic plot, such fearmongering sets a fertile ground for mobilization for a 'better future' in the following stage 3. This present-day threat scenario depicts a sealed kinship groupness of 'Arab clans' in which political authority and legitimacy remains with the (male) extended family head and, in an imagined zero-sum game, the German state's authority is delegitimized. It is a depiction of the status quo in catastrophic decline and free-fall which mobilizes for ascent to a better future: 'our' social/moral order will decline toward zero if 'we' do not intervene decisively and conclusively in the spread of 'Arab clan enclaves' in Germany.

5.3.3. Stage 3: The Promise of Ethnonational Rebirth following 'Moral Panics' in Media Interventions into the Discourse about 'Arab Clans'

German state authorities able to respond to the present-day threats and moral panics are promised in Stage 3. They are presented as capable of undertaking effective actions to secure German generalized immigration policy through which they aim to halt the reimmigration of deported Arab clan members and further criminal activities. This post-moral-panic 'better future' portrays a desired happy ending in which this threat to 'our' society is overcome, and the social/moral order is restored in ethnonational rebirth. It entails a set of arguments under the umbrella of argumentative strategy III (policing 'Arab clans') which justify why 'we' need to unite and take immediate restrictive action against 'them'.

## 6. Conclusions

In this paper, we asked how 'Arab clans' were discursively constructed in the German media between 2010 and 2020. We identified the media's dominant narrative of ethnonational rebirth vis-à-vis 'Arab clans' which facilitates social mechanisms of particularistic and exclusionary construction into a purifying 'us' and a vilifying 'them' dichotomy.

Drawing on quantitative topic modeling, we situated the discourse about 'Arab clans' in Germany's broader discourse about 'clans'. Our interpretation of the topic models revealed that the former is the most racializing and criminalizing one, and that it is constituted of the interrelations of three macro topics: family and kinship, criminal groupness, and law and order. To illuminate what these macro topics 'mean', we conducted a qualitative in-depth discourse-historical analysis which resulted in three argumentative strategies: (I) Arab clan immigration out of control, (II) Arab clans as enclaves, and (III) policing Arab clans. Together, the quantitatively assessed macro topics and the qualitatively reconstructed discursive strategies jointly realize the German media's narrative of ethnonational rebirth calling for the policing of 'Arab clans' so as to restore a selectively idealized past social/moral order in the future.

The media intervenes in the discourse about 'Arab clans' as rescuer following a narratively constructed 'moral panic': to rebuild 'our' past pure social/moral order now in a state of decline, 'Arab clans' need intense regulation by the German state, lest white Christian Germany's fall follow. State authorities (the police, State Ministers of the Interior, politicians) and the public ("many citizens") emerge as the rescuer's (mainly male) heroic aids in this plot: either as experts who lend authority to the media savior's mission with their insights 'from the field' or as the public who has already identified 'our' present-day catastrophic decline.

Importantly, the narrative of ethnonational rebirth creates black-and-white subjectivities: a pure German hero/ine said to issue from and return to a harmonic and cohesive past freed of the villain threatening it and the threatening 'Arab clan Other'. One looming present-day threat posed by this evil antagonist to the German hero/ine alleges that the (masculine) villain encloses young and female 'Arab clan' members in sealed criminal kinship groups, thereby implying that 'we' cannot teach 'them' 'our' (allegedly) anti-sexist, anti-violent, and anti-criminal culture. Such signification practices (1) demonstrate the 'postmigrant' nature of political debates, namely, that gender politics are channeled through political debates around migration and (2) demonstrate how the media's intervention creates a racialized, criminalized, and familialized groupness through which crime and violent masculinity may be externalized from the purified German ethnonational hero/ine to an invading less-civilized Other. In a nutshell, the media's story tells a straightforward message: to prevent 'our' civic free-fall, 'we' need to police and repel the threats associated with 'the Arab clan Other', thereby manifesting the promise of the celebratory return to 'our true' peaceful and cohesive past.

Future research might explore the discursive connections between Islam and 'Arab clans', something implicitly present in this paper (i.e., frequent references to 'Arab', Arab first names such as Ali and Arafat, and to alleged parallel societies as symbols for 'reli-

gious culture') could be analyzed more explicitly in a paper tailored toward that specific research question. Another promising avenue for research might be the exploration of how organized crime is racialized in the media via recourse to temporal dynamics in answer to the question of whether specific key events trigger discursive changes, i.e., did discursive changes after Germany's 'summer of migration' in 2015 change media representations of 'Arab clans'? Furthermore, in such studies, the role of gender could be important to analyze (Yurdakul and Korteweg 2021). A second line of research might conduct cross-country research in media discourses on 'Arab clans' in order to find out if, and when, the emplotment changes throughout European countries and beyond.

**Author Contributions:** Conceptualization, Ö.Ö., B.N. and G.Y.; methodology, Ö.Ö. and B.N.; software, B.N.; validation, Ö.Ö., B.N. and G.Y.; formal analysis, Ö.Ö. and B.N.; investigation, Ö.Ö., B.N. and G.Y.; resources, Ö.Ö., B.N. and G.Y.; data curation, B.N.; writing—original draft preparation, Ö.Ö., B.N. and G.Y.; writing—review and editing, Ö.Ö., B.N. and G.Y.; visualization, Ö.Ö. and B.N.; supervision, Ö.Ö., B.N. and G.Y.; project administration, Ö.Ö., B.N. and G.Y.; funding acquisition, G.Y. All authors have read and agreed to the published version of the manuscript.

**Funding:** The article processing charge was funded by the Deutsche Forschungsgemeinschaft (DFG, German Research Foundation)—491192747 and the Open Access Publication Fund of Humboldt-Universität zu Berlin. We are grateful to Berlin University Alliance, Humboldt-Universität zu Berlin for supporting the research and publication.

**Institutional Review Board Statement:** Not applicable.

**Informed Consent Statement:** Not applicable.

**Data Availability Statement:** The included data is available at https://box.hu-berlin.de/d/5ecf6659eb934118ae4f/, accessed on 22 January 2023 (Password: clancrim2023).

**Acknowledgments:** We would like to thank both anonymous reviewers, Anna Korteweg, Erik Bleich, Bernhard Forchtner, Karen Farquharson and Ashleigh Haw for their valuable feedback to earlier versions of the manuscript. We would like to further thank Tamar Sarkissian for her great technical assistance.

**Conflicts of Interest:** The funders had no role in the design of the study; in the collection, analyses, or interpretation of data; in the writing of the manuscript; or in the decision to publish the results.

## Notes

[1] The asterisk indicates the inclusion of an unlimited number of additional characters to the search word root of the search term. Hence, derivatives (e.g., 'clans') or composite terms (e.g., 'clancriminality') are included in the search.

[2] The most salient terms of each topic in the 'Arab clans' cluster were used for this search.
Topic 2: kriminalität ('criminality'), berlin, clans, kriminelle ('criminals'), organisierte ('organised');
Topic 4: miri, abschiebung ('deportation'), ibrahim, libanon ('Lebanon'), asylantrag ('application for asylum');
Topic 6: prozess ('trial'), angeklagten ('accused'), gericht ('court');
Topic 9: bushido, arafat, rapper, yasser, geschäftspartner ('business partner');
Topic 16: reul, nrw, clankriminalität ('clan criminality'), cdu;
Topic 17: polizei ('police'), festgenommen ('arrested'), verletzt ('harmed'), beamten ('officers'), streit ('argument'). The entire topic model output and the interactive intertopic distance map are available in https://box.hu-berlin.de/d/5ecf6659eb934118ae4f/, accessed on 22 January 2023 (Password: clancrim2023).

[3] Translation:
Car industry: staatsanwaltschaft = state attorney, milliarden = billions,
MENA: tunesien = Tunisia, Lybien = Libya, Syrien = Syria, regierung = government, präsident = president,
Arab clans: großfamilien = extended families, polizei = police, verbrechen = crimes

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
