# Peer review of "The ‘Arab Clans’ Discourse: Narrating Racialization, Kinship, and Crime in the German Media"

_socsci, doi:10.3390/socsci12020104_

Round 1

Reviewer 1 Report

Dear co-authors,

Warm congratulations on your excellent work which highlights the important role played by the media in the issue of societies' perspective on the communities resulting from modern migration flows.

Here are our Remarks about your work:

1.      The work has a perfect structure, grammar and clarity.

2.      The way your data is collected should be presented more.

3.      The images and graphics should be improved.

4.      Methodologically, the comparison of data on modern migration flows in relation to previous years can lead to erroneous judgments and comparisons. Although your work is excellent in everything, it could be seriously upgraded if you compared your data with data from of the same time period from other European countries that welcomed Arab populations, such as Greece, Italy, Spain, Sweden and the Netherlands.

Author Response

Thank you so much for your constructive feedback. We appreciate it a lot.

Following your comment regarding more transparency in terms of data collection, we added half a paragraph on data collection which indeed was a bit thin before. We further added some more detail to the collected date (e.g., the media outlets' political leaning) to disclose our data collection strategy and the nature of the media outlets we derived our data from.

Your other comment on the comparative aspect is of course right. However, this detailed analysis of the German media discourse about 'Arab clans' did not allow for a European comparison. As mentioned in the future outlook (final paragraph of the Conclusion), a cross-national comparison between Arab receiving European countries is essential to understand transnational similarities and national peculiarities within Europe. In fact, we were collaborating with colleagues in Australia for a cross-national comparison even beyond Europe (which is an interesting comparison because it contrasts Germany's media discourse with that of a settler colonial society). So absolutely yes, future research should definitely address that based on our national single case study. However, the depth of our analysis of the German media discourse did not allow for a cross-national comparison in this manuscript.

We hope that we were able to address your concerns.  

Reviewer 2 Report

This is an extremely well researched and written manuscript. It has a good research design, and presents an innovative, highly relevant and a timely academic study of contemporary racialized media narratives. 

With all of the merits of this submission, I do believe it is possible to 'squeeze' a little more value from this study. For the purposes of achieving this end, I would suggest that the author(s) may consider introducing the theory of 'moral panic' as an additional means of drawing even more interpretative value. Given the content of the submission, it does seem that the German media are attempting to engineer a mediatized moral panic. 

Author Response

Dear Reviewer, many thanks for your positive and constructive evaluation of our manuscript. We appreciate your kindness and constructive tone a lot. 

We absolutely agree with your point on moral panics and are thankful that you pointed that out. Yes, the German media's description of the status quo vis-à-vis 'Arab clans' manufactures moral panics and singles 'Arab clans' out as so-called folk devils. 

In fact, that concept was integrated in the manuscript in the past and then we dropped it for reasons we can not recall anymore. In any case, we integrated quotes by and references to Cohen and Hall in the Introduction and following Chapters of the manuscript. We understand moral panics as a narratively constructed status description in the middle part of a narrative (in the cases of romance and comedy) which we now stressed in the manuscript. Empirically, we of course find moral panics in the racialized and criminalized discursive construction of 'Arab clans' for which we also added mentions in the empirical section and the Conclusion of the manuscript. 

We hope that this form of integration of moral panics addresses your concerns.